# Noncompliance with Hypertension Treatment and Related Factors among Kumamoto Earthquake Victims Who Experienced the COVID-19 Pandemic during Postearthquake Recovery Period

**DOI:** 10.3390/ijerph20065203

**Published:** 2023-03-15

**Authors:** Ayako Ide-Okochi, Mu He, Hiroshi Murayama, Tomonori Samiso, Naoki Yoshinaga

**Affiliations:** 1Graduate School of Health Sciences, Kumamoto University, Kumamoto City 862-0976, Japan; 2Graduate School of Health Sciences Education, Kumamoto University, Kumamoto City 862-0976, Japan; 3Research Team for Social Participation and Community Health, Tokyo Metropolitan Institute of Gerontology, Tokyo 173-0015, Japan; 4Health and Welfare Policy Division, Health and Welfare Bureau, Kumamoto City 860-0808, Japan; 5School of Nursing, Faculty of Medicine, University of Miyazaki, Miyazaki City 889-1692, Japan

**Keywords:** earthquake, hypertension, treatment noncompliance, COVID-19, income, self-rated health, housing

## Abstract

Survivors of the Kumamoto earthquake of 2016 experienced the coronavirus disease (COVID-19) outbreak while carrying additional burdens that might bring inadequate coping. This cross-sectional survey aimed to identify untreated and interrupted consultations among those with hypertension and related factors and to identify the disaster’s long-term effects. Of the 19,212 earthquake survivors who had moved to permanent housing, 7367 (4196 women and 3171 men, mean age 61.8 ± 17.3 years) completed a self-administered questionnaire. The prevalence of hypertension was 41.4%. The results of the logistic regression analysis with the significant independent variables in the bivariate analysis were: reduced income due to COVID-19 (AOR = 3.23, 95%CI = 2.27–4.58) and poor self-rated health (AOR = 2.49, 95%CI = 1.72–3.61) were associated with a risk of untreated or discontinued treatment. Moreover, living in rental, public or restoration public housing was also significantly associated with a higher risk of hypertension noncompliance (AOR = 1.92, 95%CI = 1.20–3.07; AOR = 2.47, 95%CI = 1.38–4.42; AOR = 4.12, 95%CI = 1.14–14.90). These results suggest that changes due to COVID-19, the extent of self-rated health and the type of permanent housing influence the hypertension consulting behaviour of earthquake survivors during recovery. It is crucial to implement long-term public support for the mental health, income and housing concerns of the survivors.

## 1. Introduction

The Kumamoto earthquake, one of the largest recent earthquakes in southwestern Japan, caused seven M5.4–7.3 earthquakes over 3 days from 14–16 April 2016 [1]. Many surface ruptures occurred, destroying or partially damaging more than 188,000 houses in Kumamoto Prefecture [2,3]. Aftershocks felt by the human body continued [4], and there were more than 4364 tremors for 15 months after the earthquake [5]. Severe shaking and prolonged aftershocks also affected the health of survivors. The number of certified earthquake-related deaths due to indirect causes, such as venous thrombosis, is four times higher than that directly caused by an earthquake [6]. In Kumamoto City, the prefectural capital, up to 110,000 people out of approximately 740,000 were evacuated due to collapsed buildings. Evacuees who were unable to rebuild their homes moved into temporary housing. Subsequently, most of the affected Kumamoto City residents had moved into permanent housing by 2020. However, moving to permanent housing did not warrant easy living free from health concerns [7]. Particular attention should be paid to the risk of hypertension because hypertension-related diseases may develop immediately after a disaster. According to a review, cardiovascular diseases (CVDs) increased after the Great East Japan Earthquake (GEJE) [8]. Additionally, a previous study in Sendai, an area that was devastated during the GEJE, reported that systolic blood pressure was significantly elevated immediately after the earthquake [9]. Whilst a decrease in blood pressure was reported over a month afterwards [9], the risk continues until the environment improves [10]. The stress from the earthquakes, as well as aftershocks, house damage and obesity due to carbohydrate- and salt-rich food in shelters, are all risk factors for hypertension and CVD [8,11,12]. Furthermore, 1 year after the GEJE, withdrawal from hypertension treatment (HTTx) increased among the tsunami-ridden coastal area residents experiencing psychological distress [13]. Therefore, there is a need to clarify the medium- and long-term consulting behaviours of Kumamoto earthquake victims with hypertension. However, previous research has not elucidated the middle- and long-term effects on survivors’ behaviours of seeking medical care.

Furthermore, Kumamoto earthquake survivors incidentally suffered calamitous events during the recovery period. The final stage of settlement from temporary housing occurred during the coronavirus disease (COVID-19) pandemic. In Japan, the first wave peaked in April 2020, which followed the initial state of emergency in which the government requested residents to voluntarily take preventive actions, including social-distancing measures. Following that, Japan experienced seven more infection waves and declarations of varying levels of states of emergency, with 31,980,878 cases of COVID-19 and 65,043 reported deaths as of 22 January 2023 [14]. Such behavioural restrictions can affect health, safety, and well-being and cause mental distress and unhealthy behaviour [15]. In Japan, the COVID-19 disaster brought a 32% decrease in CVD-related hospitalisations from January 2017 to November 2020 [16]. Japan uses a universal healthcare system, but patients must pay 10–30% of their total medical fees relative to their age and socioeconomic status. In addition to COVID-19-related factors, prior to the pandemic, there were concerns that people affected by the Kumamoto earthquake might be reluctant to consult a doctor [17]. The prefectural association of doctors and dentists surveyed its members and found that 46% of doctors and dentists felt that some patients had reduced or suspended the number of medical examinations since October 2017, when the medical fee waiver for earthquake victims in Kumamoto ended [18]. However, we have little information about noncompliance with HTTx among earthquake survivors who experienced a pandemic during the recovery period.

To prevent deterioration in the affected population’s health, it is necessary to identify the factors associated with medical care noncompliance. However, factors associated with noncompliance among earthquake survivors remain largely unknown. One factor, high psychological distress, was significantly associated with the discontinuation of HTTx among survivors 1 year after the GEJE. Generally, nonadherence to HTTx is related to demographic, socioeconomic, concomitant medical–behavioural conditions, therapy-related, healthcare team and system-related, and patient factors [19]. Nonetheless, few studies have adequately clarified the factors associated with nonadherence to medical treatment for survivors who have experienced double disasters.

Therefore, this study aimed to clarify the circumstances and related factors regarding HTTx noncompliance among the affected population 4 years after the Kumamoto earthquake and to obtain suggestions for prevention strategies for health deterioration. We hypothesised that the risk of missed or interrupted consultations is related to the patients’ attributes, socioeconomic difficulties, and psychological aspects.

## 2. Materials and Methods

### 2.1. Participants

All adult survivors of the 2016 Kumamoto earthquake (19,212 individuals) who left temporary housing in December 2019 and were living in Kumamoto City in 2020 participated in this survey. Using the city’s Kumamoto Earthquake Affected Persons Register, Kumamoto City administrative officials extracted the address of the head of the household and the number of family members aged ≥18 years. The total population of Kumamoto City in 2020 was 738,567 (348,684 males and 389,883 females), with an elderly population (aged > 65 years) of 196,435 (31.9%) [20]. This cross-sectional study used a 49-item self-administered questionnaire. This questionnaire requested information regarding the presence or absence of diagnosed diseases, treatment status, mental health risks [7], living conditions, social relationships and socioeconomic factors related to COVID-19. Apart from the regular survey conducted by the municipality, this survey was carried out in collaboration with the university to acquire further detailed information before the 5-year commemoration of the disaster. Approval was obtained from the university’s ethics committee on 4 June 2020 (approval no. 1940).

The questionnaire, instructions and a return envelope for the number of adult family members were mailed to 11,479 households. The questionnaire explained the reasons for selection, ensured voluntary participation with no penalty for refusal due to possible discomfort in remembering the disaster, ensured protection of personal data and contained information on the publication of the survey results and contact details. The researchers also explained that returning the questionnaire form constituted consent. Each family member completed and mailed the questionnaire individually. The data collection period was from July to December 2020. This period coincided with the second and third waves of COVID-19 [16].

Questionnaires were returned by 9409 participants, producing a 49.0% return rate of the questionnaire. We excluded 130 questionnaires returned without answers, 313 questionnaires with more than 95% unanswered questions and 1599 questionnaires with unanswered questions regarding hypertension. Among these 7367 respondents, our analysis used data from 763 persons who completed the questions regarding the hypertension treatment (HTTx) status (Figure 1). The return of written questionnaires was considered consent [7].

### 2.2. Hypertension Treatment Noncompliance

The participants self-reported whether or not they had a previous hypertension diagnosis. They were instructed to tick none if treatment had been completed. Those who responded ‘yes’ to the diagnosis of hypertension were then prompted to answer questions on their treatment status. Respondents chose between ‘on treatment’, ‘untreated’ and ‘interrupted treatment’. Those currently on medication, treatment or following doctor’s follow-up instructions were instructed to tick ‘on treatment’. Furthermore, those who marked ‘untreated/interrupted treatment’ were asked to explain with multiple answers. Both ‘untreated’ and ‘interrupted treatment’ were defined as treatment noncompliance.

### 2.3. Other Variables

#### 2.3.1. Attributes

Attributes analysed included sex, age and whether they lived with someone. Age was categorised into two groups: 18–64 and 65+, defining the elderly in Japan.

#### 2.3.2. Housing Conditions

Kumamoto City built restoration public housing (RPH) as permanent housing for people who lost their homes during the Kumamoto earthquake and had difficulty rebuilding or renting on their own. RPH has a higher proportion of earthquake victims than other types of public housing, and their isolation is a concern. For example, 10 years after the GEJE, a total of 614 lonely deaths occurred in Iwate, Miyagi and Fukushima prefectures, and 341 of such deaths were RPH residents [21]. The questionnaire asked about their living situation after leaving temporary housing (resettlement) and whether they had changed their primary school district due to resettlement. A change in the school district also indicated relocation, a risk factor for mental health problems. For example, survivors who moved due to the 2011 Christchurch earthquake had a significantly higher risk of seeking treatment for psychological symptoms than those who did not change their predisaster address [22].

#### 2.3.3. Relations with Society

Loneliness and isolation are estimated as health risks comparable to the harmful effects of tobacco [23]. A decade after the GEJE, poor subjective health was related to the fear of loneliness [24]. Participants were asked whether they had ever felt isolated from others on a four-point scale (not at all, rarely, sometimes and constantly); ‘not at all’ and ‘rarely’ were categorised as a state of no loneliness and ‘sometimes’ and ‘constantly’ as a state of loneliness. Furthermore, regarding the status of social participation, respondents indicated their involvement in events and social gatherings in their local area by selecting ‘I participate’, ‘I don’t participate’ or ‘I don’t have much information’.

#### 2.3.4. Health Conditions Other than Hypertension

Self-rated health is an overall evaluation index of subjective health status. In the Comprehensive Survey of Living Conditions and National Health and Nutrition Survey, the question ‘How is your current health condition?’ allows the following options as answers: 1, very good; 2, good; 3, somewhat good; 4, somewhat bad; 5, bad; or 6, very bad [25]. Self-rated health is a subjective way of evaluating health, as it is relative to the health goals set by each individual. Additionally, this indicator predicts life expectancy and disease occurrence in Japan, Europe and the United States [26].

#### 2.3.5. Changes Due to COVID-19

We asked about changes in their incomes due to COVID-19. In the analysis, ‘considerably decreased’ and ‘somewhat decreased’ were categorised as a decrease and ‘no decrease’, as stated.

### 2.4. Data Analysis

We calculated the descriptive statistics for all variables. Next, we conducted a test of independence (a chi-square (χ^2^) test) to examine the association between HTTx noncompliance and the variables of attributes, residence, social relationships, self-rated health and COVID-19 impact. Then, we conducted binary logistic regression analysis to examine the factors influencing refraining from seeing a doctor. The model was adjusted for sex, age, presence of a cohabitant, permanent residence category, change in school district (change of residence), community participation, loneliness, subjective health and change in income due to COVID-19. Adjusted odds ratios (AORs) and 95% confidence intervals (95% CIs) were used to examine the results. Before logistic regression analysis, we checked multicollinearity between independent variables to ensure that they did not exhibit a variance inflation factor (VIF) of >10. Variables were selected using the forced assignment method. We assessed the goodness of fit of the model using the Hosmer–Lemeshow test. For each variable, we excluded missing values from the analysis. Statistical analyses were conducted using IBM SPSS 28 for Windows (IBMCorp., Armonk, NY, USA), with a statistical significance level set at α = 0.05.

## 3. Results

Among the 7367 (4196 women and 3171 men, mean age 61.8 ± 17.3 years) respondents who answered the question regarding the presence of hypertension, 3053 (41.4%) answered yes.

Table 1 shows the demographic characteristics of the analysed respondents (*n* = 763), including women (57.0%) and men (43.0%), with age group (51.0% aged 65 and over, mean age 62.69 ± 15.04), current residence (48.9% lived in owner-occupied housing) and relocation (42.7% changed their school district when moving to permanent housing). Additionally, 18.5% participated in community activities. A total of 35.5% also reported feeling lonely.

Predominant reasons reported for untreated/interrupted treatment were financial burden, not feeling the need to see a doctor and lack of time (Figure 2).

Table 2 shows the results of the χ2 test between the independent variables and the presence or absence of refraining from HTTx. A higher proportion of hypertension treatment noncompliance was evident in the younger survivors, those with lower self-rated health, those who changed school districts, those in housing other than a self-owned house, those experiencing loneliness and those with decreased income due to COVID-19.

Table 3 shows the results of the binary logistic regression analysis with HTTx noncompliance as the dependent variable and attributes, social relationships, subjective health and change due to COVID-19 as the independent variables. Logistic regression analysis indicated that age (AOR = 0.97, 95% CI = 0.96–0.99), rental housing (AOR = 1.92, 95% CI = 1.20–3.07), public housing (AOR = 2.47, 95% CI = 1.38–4.42), RPH (AOR = 4.12, 95% Cl = 1.14–14.90), poor self-rated health (AOR = 2.49, 95% CI = 1.72–3.61) and reduced income due to COVID-19 (AOR = 3.23, 95% CI = 2.27–4.58) were associated with HTTx noncompliance. The result of the Hosmer–Lemeshow test was *p* ≥ 0.05, indicating that the fitness of the model was acceptable.

## 4. Discussion

To the best of our knowledge, this is the first study to assess HTTx noncompliance in earthquake survivors 4 years after a major earthquake of magnitude >7 and to identify associated factors, including COVID-19-related changes. The results of this study provide guidance for the long-term prognosis of hypertension in earthquake victims and for secondary prevention measures.

The prevalence of hypertension remains high. In 2019, the prevalence of hypertension among Japanese adults was 47.7% (men: 56.7%; women: 41.7%) [27]. Incidentally, in the census above, the prevalence of hypertension was defined as systolic blood pressure of 140 mmHg or more, diastolic blood pressure of 90 mmHg or more or those taking blood-pressure-lowering medication [27]. In this study, hypertension was self-reported by participants. However, the prevalence of hypertension among the participants also appears to reflect trends in Japan.

### 4.1. Factors Related to HTTx Noncompliance

Hypertension is a major risk factor for future CVD. Out of the estimated 43 million patients with hypertension in Japan, most cases (31 million, 72%) were poorly controlled [28]. Factors such as rental housing, public housing, RPH, poor subjective health and reduced income due to COVID-19 were associated with withholding HTTx. A discussion of these factors is provided below.

#### 4.1.1. Types of Permanent Dwellings

Not owning one’s current residence was significantly associated with noncompliance with HTTx. Existing studies have examined the association between housing type and mental health among GEJE survivors. Two and a half years after the GEJE, adjusted rate ratios for depressive symptoms among older survivors were almost twice as high among residents in prefabricated temporary housing compared with those living in other housing types [29]. A survey conducted in Fukushima Prefecture reported that subjective well-being was lower in RPH compared with non-RPH residents; however, it was not statistically significant after more than 7 years after the GEJE [30]. Thus, previous research related to GEJE implied the possible association between housing type and survivors’ perceived health. However, there has been scarce research that illuminated the relationship between noncompliance and housing types among earthquake survivors. Hence, this research is able to contribute to academic societies and policymakers by proving the high OR (4.12) of HTTx noncompliance among RPH residents.

One factor that influences long-term adherence among survivors of chronic illnesses is socioeconomic context. A meta-analysis reported that socioeconomic status (SES) could be an important determinant of nonadherence to antihypertensive drugs, although the examined cohort studies indicated inconsistent findings [31]. Survivors who find it difficult to rebuild their homes and face financial hardship are likely to move into rented, public or public disaster housing. However, the relationship between the type of permanent housing after moving out of temporary housing and HTTx noncompliance has not been investigated adequately. This study is one of the few to identify the association between housing type and HTTx noncompliance during the recovery period after leaving temporary housing. Long-term studies examining the association between the type of permanent housing after leaving temporary housing and long-term adherence to treatment of chronic disease are warranted.

#### 4.1.2. Self-Rated Health

The risk of withholding HTTx was higher among those with poor subjective health status. Self-rated health may be associated with treatment compliance by patients. A study of Chinese adult hypertensive patients found a significant association between good self-rated health and controlled blood pressure [32]. In a survey of hypertensive patients in health maintenance organisations and veterans affairs medical centres in the USA, self-rated depressive symptoms and noncompliance were significantly associated [33]. In addition, a cohort study in Sweden showed that health self-rated as good was significantly associated with reduced mortality in hypertensive men (adjusted relative risk = 0.56, 95% CI = 0.34 to 0.91) [34]. Subjective health and adherence to treatment have also been reportedly associated among earthquake victims. A previous study found significantly higher OR (4.0) for HTTx withholding for those with high psychological distress (K6 ≥ 13) compared with those with relatively low psychological distress (K6 < 13) in a coastal affected area 1 year after GEJE [13]. Hence, it is likely that earthquake survivors with hypertension with poor subjective health perceptions were, therefore, at a higher risk of treatment noncompliance. Future long-term follow-up of the prognosis of HTTx among individuals with poor subjective health outcomes is needed.

#### 4.1.3. Change Due to the COVID-19 Pandemic

In this study, we were able to show that income loss due to COVID-19 was significantly associated with HTTx noncompliance. In the review above by Canadian researchers, the medium- to long-term impact of COVID-19 on cardiac risk factor management may include reduced adherence to CV risk reduction therapies and reduced primary and secondary prevention of cardiovascular events due to delayed outpatient visits [35]. According to this review, reduced income and unemployment are among the possible effects of the COVID-19 pandemic on subsequent cardiovascular mortality and morbidity. In general, financial losses and difficulties in accessing healthcare affected the long-term adherence of chronically ill patients [19]. However, to our knowledge, few studies have demonstrated an association between the economic impact of COVID-19 and nonadherence to HTTx among earthquake survivors.

The association between delayed or lost cardiac healthcare as an indirect effect of COVID-19 and socioeconomic loss due to COVID-19 is only speculative [35]. This study’s participants were victims of a double disaster, having been exposed to the COVID-19 pandemic in the middle of their recovery from the earthquake. This study showed that reduced income due to COVID-19 might contribute to HTTx nonadherence among earthquake survivors. Therefore, the outbreak of the pandemic and the associated reduction in income can be considered an additional burden following the Kumamoto earthquake. It may be necessary to recognise this new burden caused by COVID-19 and, therefore, support survivors to prevent negative long-term health consequences.

### 4.2. Limitations and Significance of the Study

There are several limitations that should be taken into consideration when interpreting the current findings. Firstly, this study was limited to victims of the Kumamoto earthquake, and the results obtained may not be applicable to all earthquake victims. Secondly, as this was a cross-sectional study, it was not possible to prove a causal relationship. Therefore, longitudinal studies are required to explain causality. Additionally, although various interactions are possible, there are also few empirical studies on the effects of COVID-19 during the reconstruction period on earthquake victims. Thirdly, important determinants other than the variables included in this study may exist. Fourthly, it is unclear whether the impact of risk factors is specific to survivors in the postdisaster recovery phase. Future research needs to make comparisons with the general population who are not disaster victims. Finally, reports on hypertension diagnosis and HTTx were self-reported, and measurement bias cannot be ruled out. The validity and accuracy of the items need to be examined in the future. Future research should include a more accurate analysis using national databases on healthcare and care costs. Despite the above limitations, this study was able to identify the factors associated with HTTx noncompliance among survivors who moved from temporary to permanent housing 4 years after the Kumamoto earthquake and provides insights for future support planning.

## 5. Conclusions

Four years after the Kumamoto earthquake, the OR of RPH was the highest among the factors associated with HTTx noncompliance among the affected population. Lower incomes due to COVID-19 were also significantly associated with inappropriate treatment-receiving behaviours. During the recovery period from the effects of the earthquake, living in nonowner-occupied dwellings, such as rented, public or RPH was associated with HTTx noncompliance. In addition, economic changes due to the COVID-19 pandemic could be an additional burden and were associated with HTTx noncompliance. Research and long-term support are essential to prevent continuing adverse consequences for the victims of the Kumamoto earthquake.

## Figures and Tables

**Figure 1 ijerph-20-05203-f001:**
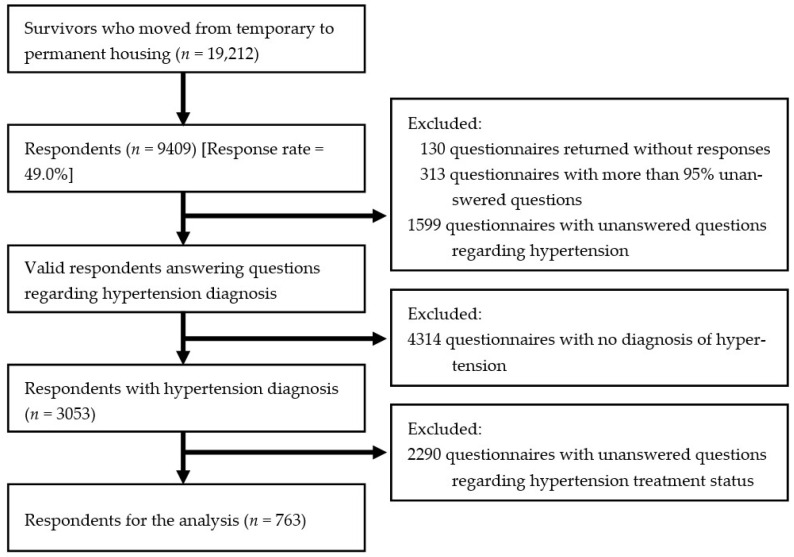
Flow chart summarising the number of respondents included in the analysis (*n* = 763).

**Figure 2 ijerph-20-05203-f002:**
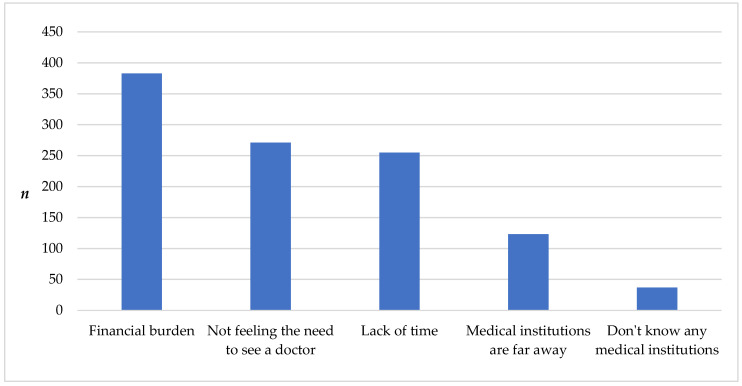
Reasons for HTTx noncompliance (multiple answers).

**Table 1 ijerph-20-05203-t001:** The demographic characteristics of the analysed respondents.

			*n* = 763
		*n*	%
Sex	Male	328	43.0
	Female	435	57.0
Age	18–64 years old	374	49.0
	65+ years old	389	51.0
Interruption of treatment	None	525	68.8
	Yes	238	31.2
Current residence	Owned house	373	48.9
	Houses for rent	240	31.5
	Public housing	114	14.9
	RPH	12	1.6
	Other	24	3.1
Cohabitant	None	199	26.1
	Yes	564	73.9
Self-rated health	Not healthy	274	35.9
	Healthy	489	64.1
Loneliness	None **	492	64.5
	Yes	271	35.5
Community participation	No information of such events	108	14.2
	None	514	67.4
	Yes	141	18.5
Change of residential school district	None	437	57.3
	Yes	326	42.7
Decrease in income due to COVID-19	None ***	426	55.8
	Yes	337	44.2

Abbreviations: COVID-19, coronavirus disease 2019; RPH, restoration public housing. ** ‘None’ for loneliness includes never or hardly ever; ‘Yes’ for sometimes or always. *** ‘None’ for decrease in income includes no change; ‘Yes’ for both large and small decreases.

**Table 2 ijerph-20-05203-t002:** Participant characteristics by hypertension treatment (HTTx) noncompliance.

							*n* = 763
		HTTx Noncompliance					
		Applicable (Untreated or Interrupted Treatment)		Not Applicable (under Treatment)			
		*n* = 238		*n* = 525		*p*-Value	Cramér’s V
		*n*	%	*n*	%		
Sex						0.914	0.040
	Male	103	43.3	225	42.9		
	Female	135	56.7	300	57.1		
Age		143	60.1	231	44.0	<0.001	0.149
	18–64 years old	95	39.9	294	56.0		
	65+ years old						
Cohabitant						0.052	0.070
	None	73	30.7	126	24.0		
	Yes	165	69.3	399	76.0		
Self-rated health						<0.001	0.221
	Not healthy	123	51.7	151	28.8		
	Healthy	115	48.3	374	71.2		
Current residence						<0.001	0.216
	Owned house	79	33.2	294	56.0		
	Houses for rent	94	39.5	146	27.8		
	Public housing	50	21.0	64	12.2		
	RPH	6	2.5	6	1.1		
	Other	9	3.8	15	2.9		
Change of residential school district						0.002	0.110
	None	117	49.2	320	61.0		
	Yes	121	50.8	205	39.0		
Loneliness						<0.001	0.156
	None	127	53.4	365	69.5		
	Yes	111	46.6	160	30.5		
Community participation						0.734	0.028
	No information of such events	37	15.5	71	13.5		
	None	159	66.8	355	67.6		
	Yes	42	17.6	99	18.9		
Decrease in income due to COVID-19						<0.001	0.290
	None	82	34.5	344	65.5		
	Yes	156	65.5	181	34.5		

The chi-square test of independence (χ^2^ test) was performed.

**Table 3 ijerph-20-05203-t003:** Related factors of hypertension treatment (HTTx) noncompliance (binary logistic regression analysis).

			*n* = 763
		HTTx Noncompliance	
		AOR	95% CI
Sex (ref: male)			
	Female	1.09	0.76–1.55
Age (ref: 18–64 years old)			
	65 years and older	0.97	0.96–0.99
Cohabitant (ref: yes)			
	No	1.08	0.72–1.63
Current residence (ref: owned house)			
	Houses for rent	1.92	1.20–3.07
	Public housing	2.47	1.38–4.42
	RPH	4.12	1.14–14.90
	Other	2.33	0.89–6.10
Change of residential school district (ref: none)			
	Yes	0.87	0.57–1.33
Loneliness (ref: none)			
	Yes	1.43	0.99–2.06
Community participation (ref: yes)			
	No information of such events	0.60	0.32–1.12
	None	0.70	0.44–1.11
Self-rated health (ref: healthy)			
	Not healthy	2.49	1.72–3.61
Decrease in income due to COVID-19 (ref: none)			
	Yes	3.23	2.27–4.58

Performed binary logistic regression analysis. Variable selection was forced entry. *p* ≥ 0.05 for Hosmer–Lemeshow test results for logistic regression analysis. Abbreviations: AOR, adjusted odds ratio, 95% CI 95%, 95% confidence interval.

## Data Availability

The data presented in this study are available upon request from the corresponding author.

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
