# Peer review of "Noncompliance with Hypertension Treatment and Related Factors among Kumamoto Earthquake Victims Who Experienced the COVID-19 Pandemic during Postearthquake Recovery Period"

_ijerph, 2023, doi:10.3390/ijerph20065203_

Round 1
Reviewer 1 Report
Abstract
1. Survey details should be included under methods including study design (e.g. cross-sectional) and type of questionnaire used.
2. Findings from the bivariate analyses have not been reported.
3. When presenting the results of the logistic regression model, reference category should be mentioned e.g. compared with. Also, need to make sure that the reported ORs are adjusted not unadjusted.
4. Under recommendation, it is not clear what type of support was recommended e.g. financial, mental health etc and who will be responsible for this type of support.
Material and methods
1. More information about 49-items self-administered questionnaire is required. What are the information covered in this questionnaire and how? Also, it is not mentioned how the items were measured. Need detailed explanation.
2. In the self-reported questionnaire how, the hypertension was measured. How the self-reported bias (measurement bias-yes/no) on hypertension was handled? Need some explanations.
3. Explain the hypertension questionnaire and items?
4. Data analysis section
a. Need to explain why bivariate analysis was conducted.
b. Why and how unweighted bivariate association using Chi-square test was conducted should be included in the methods section. Also, bivariate analyses should be reported first before logistic regression.
c. How the covariates were controlled in the logistic regression model was not mentioned. The OR should be adjusted for covariates and the reported values of OR should be adjusted odds ratio (AOR) not OR. These should be re-evaluated and corrected in Table 4 and in the results section.
d. Not clear how the missing values were excluded and reason behind. Which method was followed and why? This needs explanation.
Results
1. Table 1 is not required, some of the stats are repeated in Table 2. Delete Table 1. Response to hypertension data already been presented in the text. Also, not sure why age two types of age (metric and categorical) were used (Table 1 and Table 2). Need to clarify this otherwise use age as metric variable.
2. Titles of Tables 3-4 are not correct. They are not meaningful, please make sure that the title represents the results presented in these tables.
3. In the bivariate association (Table 3) why age variable was not included? Age is an important demographic variable. This should be tested under bivariate association using appropriate test statistic.
4. Include Cramer’s V in Table 3 along with p-value to show the strength of association between the variables.
5. Interpretations on the results obtained from bivariate associations presented in Table 3 were not included in the results section.
6. Report results of the bivariate association test before presentation results obtained from the logistic regression model.
Reviewer 2 Report
The authors do not adequately explain the reduction of the sample from 9409 to 763. Table 1 shows the demographics of the 9409 but then Table 2 is a subset but how is it different from the main data. Also the tables are not clearly labeled. For instance what does applicable and not applicable refer to in Table 3.
Reviewer 3 Report
My comments are provided in the attachment.

Round 2
Reviewer 1 Report
Thanks for fixing the issues in the revised version.